# The Role of Entropy in Construct Specification Equations (CSE) to Improve the Validity of Memory Tests

**DOI:** 10.3390/e23020212

**Published:** 2021-02-09

**Authors:** Jeanette Melin, Stefan Cano, Leslie Pendrill

**Affiliations:** 1Department of Measurement Science and Technology, Research Institutes of Sweden (RISE), AWL Sven Hultins Plats 5, vån 4, 412 58 Göteborg, Sweden; leslie.pendrill@ri.se; 2Modus Outcomes, Spirella Building, Letchworth Garden City SG6 4ET, UK; stefan.cano@modusoutcomes.com

**Keywords:** entropy, information, metrology, measurement system analysis, Rasch, cognition, memory, task difficulty, person ability, neurodegenerative diseases, cognitive neuroscience

## Abstract

Commonly used rating scales and tests have been found lacking reliability and validity, for example in neurodegenerative diseases studies, owing to not making recourse to the inherent ordinality of human responses, nor acknowledging the separability of person ability and item difficulty parameters according to the well-known Rasch model. Here, we adopt an information theory approach, particularly extending deployment of the classic Brillouin entropy expression when explaining the difficulty of recalling non-verbal sequences in memory tests (i.e., Corsi Block Test and Digit Span Test): a more ordered task, of less entropy, will generally be easier to perform. Construct specification equations (CSEs) as a part of a methodological development, with entropy-based variables dominating, are found experimentally to explain (*r*
$=\sqrt{R^{2}}$ = 0.98) and predict the construct of task difficulty for short-term memory tests using data from the NeuroMET (n = 88) and Gothenburg MCI (n = 257) studies. We propose entropy-based equivalence criteria, whereby different tasks (in the form of items) from different tests can be combined, enabling new memory tests to be formed by choosing a bespoke selection of items, leading to more efficient testing, improved reliability (reduced uncertainties) and validity. This provides opportunities for more practical and accurate measurement in clinical practice, research and trials.

## 1. Introduction

Worldwide, someone develops dementia every three seconds, making it one of the most pressing public health issues in modern time [1]. However, patients remain under-diagnosed or are diagnosed ‘too late’. This may be due, in part at least, to the limitations of data generated by existing, routinely used neuropsychological cognitive assessments [2,3,4]. Key drawbacks include the: (i) inherent ordinality of human responses to items assessing cognitive functioning; and (ii) lack of analytical recourse to separating the independent parameters of person ability and item difficulty. Ignoring these two main effects leads to considerable risk of incorrect conclusions based on the cognitive assessment data, impacting clinical decision-making and the assessment of the significance of correlations between cognitive functioning and brain structure and biochemistry [5]. In the present paper, we use a metrological approach to tackle these respective limitations (Figure 1).

Of note, when analyzing responses (*Y*, registered by the operator *C*), we give equal attention to variations in item (entity, *A*) values and in person (instrument, *B*) scores. This is in contrast to the more common trend in the social sciences dating back to the early 1900s (i.e., including Thurstone [9] and ‘early correlationalists’), where “most approaches to testing and construct theories focus(ed) on explaining person score variation” ([10] p. 416). Construct and measurement theories are handled with equal priority, as far as possible. In doing so, we do not share the view of others, that the instrumental task of quantification, i.e., construction of measurement devices and instruments, is “secondary” to the scientific task of discovering quantitative structure ([11] p. 359).

The concept of entropy can provide valuable insight when modeling a measurement process [12]. An information-theoretical entropy, as a measure of the amount of “useful information” in a measurement system (such as depicted in Figure 1), is analogous to a certain extent to the original entropy concept as a measure of “useful energy” in steam engines [13]. There is an extensive literature in which the validity of analogies between the entropy concept in information theory and thermodynamics is considered in depth, see for example [14]. A summary about making these analogies is given in [15], where statements can be found, such as: “Ultimately, the criticism of the link between thermodynamic entropy and information entropy is a matter of terminology, rather than substance. Neither side in the controversy will disagree on the solution to a particular thermodynamic or information-theoretic problem”. A task (entity, *A*) will be easier if there is some degree of order, i.e., less entropy. The greater the degree of order in an instrument (person, *B*; e.g., coherency), the smaller the entropy and the greater the ability to perform tasks. Additionally, both distortion and loss of information in the measurement process—that is, error and uncertainty, respectively—can be modeled in terms of increases in entropy [16], i.e., disorder. Analogies can be drawn with previously reported entropy-based measurement models, such as the treatment of machine-based signal treatment (e.g., EEG patterns in time series to detect epilepsy [17]).

Our entropy-based measurement model exploits the properties of the Rasch model [18], in which the ordinal response of a human (*Y*) in a memory test is transformed (in the process of restitution [6], *R* in Figure 1) into separate estimates of the ability, θ, of the signal analyzer (*B*) and the level, δ, of difficulty of the task (*A*) on a conjoint linear scale (Equation (3), Section 2.3). Entropy, as a concept, is used here not only to handle ordinal signals, but also explain both person ability and task difficulty, building on the pioneering work of Brillouin [19].

Following the restitution process, construct specification equations (CSE) can be formulated for each attribute of the various elements (object, *A*, instrument, *B*, etc.) of the measurement system (Figure 1), including explanation of memory task difficulty as a function of entropy. CSEs are based on our best understanding of the causality explaining each construct [20] As will be demonstrated, the concept of entropy plays a dominant role when formulating CSEs for the non-verbal memory tests examined here. For instance, one can explain the level of memory difficulty of the task (*A*), essentially as a sole function of entropy.

CSEs play an important role in the design and implementation of memory tests, since they allow: (i) the valid and reliable prediction of the difficulty of new test items (e.g., to fill measurement gaps); as well as (ii) provide a measure of the degree of equivalence between items associated with distinct tests (e.g., different cultures and languages) [20,21]. New memory tests which combine carefully selected items from diverse sources can thus be formulated. The increased number and comprehensiveness of items can be expected to lead to a reduction in measurement uncertainties much-wanted in clinical decision-making.

This paper is structured as follows. First, we introduce the memory tests and sample used in this study (Section 2.1) and the metrological quality assurance of human based measures (Section 2.2). This is followed by linking entropy to item response models (Section 2.3) and to CSEs (Section 2.4). Experimental demonstrations of how entropy can be used to: (i) explain memory task difficulty; and (ii) combine different items from different tests according to entropy-based criteria follow in Section 3. We conclude with a discussion in Section 4, explaining how the resultant improved reliability (reduced uncertainties) in memory test difficulty estimates in turn promises improved estimates of person memory abilities as key inputs to cognitive neuroscience and health care.

## 2. Materials and Methods

This section provides a description of the materials for the experimental demonstrations (Section 3) in terms of the memory tests and the samples generated by measurement. This is followed by a description of the assumptions and processes of the methods.

### 2.1. Memory Tests and Sample

The Corsi Block Test (CBT) [22] and Digit Span Test (DST) [23] are two tests, largely free of cultural and language effects, for measuring short-term memory ability. In the CBT, a person is asked to reproduce the tapping sequences (i.e., an item or a task) provided by the test-leader for a series of tasks with increasing length and difficulty. The DST similarly requires recalling tasks, but instead of tapping sequences the participants are asked to recall digit sequences. For both tests, if a person recalls a sequence correctly the observed response is scored 1; an incorrect response is scored 0. From these raw response scores the measurand is then restituted through a logistic regression of the dichotomous Rasch model (Section 2.3) [18] using the WINSTEPS ^®^ 4.3.1. software application This restitution process yields separate and linear measures for individual memory task difficulties, δ, and person memory abilities, θ subject to a good fit of the basic Rasch model [20].

The data for those memory tests stem from subjects in the project NeuroMET [24] comprising a cohort of 88 subjects with AD (n = 26), mild cognitive impairment (MCI) (n = 23) and healthy controls (HC) (n = 39). The mean age was 72 years (range 55–84 years) and 47% were women and 53% men. Data from a second version of the DST test were additionally retrieved from the GBG MCI study [25] comprising a cohort of 257 subjects with dementia (n = 18), MCI (n = 142) and HC (n = 52). The mean age was 65 years (range 40–86 years) and 49% were women and 51% men.

### 2.2. Human-Based Measures

Memory tests are a typical case of human-based measures where the observed response in a memory test can be, for example, pass or fail. Other common examples of human-based measures are questionnaires with a set of questions with rating scales such as strongly disagree to strongly agree on a 5-point Likert scale or 1–10 on numerical ratings scales (NRS) [26]. In those examples, the human responses do not generate ‘ordinary’ numbers, but instead classification scores by category (i.e., observed responses on an ordinal scale). Underlying the use of raw summed scores of such rating scales and tests are what Tukey termed ‘counted fraction’ data [27,28], where the observed response to test items [as registered by the measurement system operator (*C*) in Figure 1] are usually percentages or equivalent scores (*P_success_* in the dichotomous case, the probability of “correct” classification), bounded between lower and upper limits of 0 and 100%, and summing to 100%.

As early as 1897, Pearson [29] (p. 490) had warned of the dangers of counting fractions under such boundary conditions: “Beware of attempts to interpret correlations between ratios whose numerators and denominators contain common parts” (cited in [28] p. 315). Mathematically, in the counted fraction expression, $X_{j}\%=\frac{X_{j}}{\sum_{i}X_{i}}$, the presence of the amount *X* of component *j* (appearing in both the numerator and denominator) means that any error in *X_j_* will be correlated with the other components (and increasingly so where *X_j_* is either large or small compared with the other components), since there is the boundary condition $\underset{j}{\sum }X_{j}\%=100\%.$ Thus, counted fractions lead to a scale which becomes increasingly non-linear at either end of the response scale—high scores and low scores. This, and other potential sources of scale non-linearity (such as responder acquiescence [30]), if not recognized and compensated for, will lead to (at best) ordinal scales with a monotonic order. With unknown scale intervals even the most basic tools of statistics cannot be assumed to work. Despite the long history of counted fractions studies, there remains to this day a diversity of approaches to handling the observed data from human-based responses in for instance the medical field, health care and education.

Apart from a few exceptions (particularly the work of Stenner et al. [10,31]), the analysis of memory tests has (to date) unfortunately not accounted properly for counted fraction ordinality nor applied methods for metrological quality assurance to ensure traceability and declare measurement uncertainties. The Rasch model, dating from the 1960s, is a unique kind of item response model to handle these challenges. This is because it not only compensates for counted fraction ordinality, but also importantly provides separate estimates of each test person’s ability and each test item’s level of difficulty in a way that other ad hoc transformations traditionally employed cannot achieve. That separation, which corresponds to the measurand restitution (Figure 1), is essential for establishing metrological references for metrological traceability for human-based measures as in traditional physical metrology [7].

An early form of the Rasch model [18] posits that the odds ratio of successfully performing a task is equal to the ratio of an ability, *h*, (Rasch [17] used the person attribute ’inability’ instead, given by $h^{-1}$) to a difficulty, *k*:(1)$$\frac{P_{success}}{1-P_{success}}=\frac{h}{k}$$

The original Rasch model of Equation (1) refers to a Poisson distribution $p\left(x\right)=\frac{e^{-\lambda }\;\cdot \;\lambda^{x}}{x!};x=0.1,\ldots$, such as used traditionally in quality control as a model of the number of defects or nonconformities that occur in a unit of product [32]. The parameter λ is equal directly both to the mean and variance of the Poisson distribution and in the Rasch model $\lambda =h^{-1}\;\cdot \;k$ [18]. As will be discussed in Section 2.3, the Rasch model can in fact be viewed as an entropy-based measurement model.

### 2.3. Entropy and Item Response Models

Step-by-step in the passage of information through our prototype measurement system, as illustrated in Figure 1, the terms in the well-known conditional entropy expression are added together [33]:(2)$$H\left(Y|Z\right)=H\left(Z,Y\right)-H\left(Z\right)$$
expressing the entropy in the response (*Y*) when observing the quantity (*Z*) as the joint entropy reduced by the entropy associated with the measurement object (*A*) prior to measurement.

The link between a probabilistic description [6] and a corresponding entropy-based approach, as depicted in Figure 1, is based on the informational ‘Shannon’ entropy, *H*, of any ‘message’ of probability *P* being proportional to $log\left(P\right)$ [16]. This formulation captures the fact that the less expected a message is (i.e., smaller *P*), the greater the amount of information conveyed (‘surprisal’). Taking the logarithm also facilitates addition and subtraction of different amounts of information.

The “Shannon” entropy terms in Equation (2) can thus be expressed in terms of the probability distributions associated with each variable in the measurement process, where these probabilities are multiplied in the expression $P\left(z,y,z_{R}\right)\;=\;P\left(z\right)\;\cdot \;P\left(y|z\right)\;\cdot \;P\left(z_{R}|z,y\right)$ using the notation used by Rossi in his probabilistic model of the measurement process [6]); and *R* denotes restitution.

Equation (2) states how the amount of information changes during transmission in a measurement system. At the start of the measurement process, there is an initial ‘deficit’ in entropy (i.e., ‘surplus’ information) coming from prior knowledge, $H\left(Z;A\right)$, of the measurand (attribute, Z, of entity *A*), again using a notation analogous to that used by Rossi in a probabilistic model of the measurement process [6]. Losses and distortions $H\left(Y,Z\right)$ from imperfections in the measurement process increase the entropy, leading finally to a posterior distribution (*Q*) with entropy $H\left(Y|Z\right)$ as the result of the measurement process.

As will be described in detail in Section 2.4 and Section 3, entropy can form the basis of a semantic description of the object (*A*) construct task difficulty through formulation of the entity entropy $H\left(Z\right)$ in terms of our understanding of what determines task difficulty. Entropy, *H*, can also be connected to the Rasch model (Equation (1)) allowing the probabilistic response of the instrument (*B*) to be related to task difficulty and person ability with the expression:(3)$$log\left(\frac{P_{success}}{1\;-\;P_{success}}\right)\;=\;log\left(\lambda \right)\;=\;log\left(k\right)\;-\;log\left(h\right)$$
where there is a dichotomous outcome inherent in a Poisson distribution [34]. Following Rasch [18], the Poisson distribution factor λ, includes both *h* and *k* in Equation (1) as the probabilistic ability and difficulty factors, respectively. The test person ability, $\theta =log\left(h\right)$, and task difficulty, $\delta =log\left(k\right)$, are essentially “surprisal”, entropy-based terms, and can be evaluated by logistic regression of Equation (3) to the raw score data (exemplified in Figure 2) in terms of the probabilities *q* (*P_success_*) of how well a measurement system, with a person as the measurement instrument, performs [5].

To pick up on the issue of non-linearity and ordinality due to counted fractions (Section 2.2), typically, an S- or ogive-curve (Figure 2a,b) is expected from Equation (3) when correlating observed responses $P_{success}$ with test item difficulty and person ability, respectively, across the combined CBT and DST tests investigated in Section 3.4, for example. If one does not (as in CTT) recognize and compensate for this ordinality in the observed responses, and do not separate item difficulty from person ability, then both item difficulties and person abilities will be underestimated or overestimated at either scale extreme (Section 2.2). These effects have to be investigated, however large or small they might turn out to be. A well-designed test, with good targeting, where the span of person abilities matches well (both in terms of centering and width) the corresponding range of task difficulty, will in general have small non-linearities. Note that, in any case, the Rasch model is essential for metrological quality assurance since it provides, through measurand restitution, separate values for task difficulty and person ability.

As in any measurement system, the instrument (*B* in Figure 1) has a certain sensitivity, $K=\frac{\partial P_{success}}{\partial \delta }$, which gives the ratio of the response to a stimulus, in the present case how much each test person responds to a task of a certain difficulty, δ. While the basic Rasch model (Equation (3)) assumes *K* = 1, in some cases an additional discrimination factor, *ρ*, of the person responding is included on the RHS of Equation (3) as the term $\rho \;\cdot \;\left(\theta -\delta \right)$ in a so-called two-parameter item response model. This captures the possibility that sensitivity can vary from person to person and item to item, for instance because of illness or emotional reactions [28]. An entropy-based model of item response can then include, apart from task difficulty δ and person ability θ, an additional entropy contribution. This can both contribute as a constant offset (intercept) in the overall entropy with the term $H\left(Y,Z\right)\;\sim \;ln\left(\rho \right)$, as well as to uncertainties in the Rasch model estimates of task difficulty δ and person ability θ, to account for potential variations in discrimination from person to person and/or item to item. The latter uncertainties can be estimated from the entropy relation $H\left(Y,Z\right)\;\sim \;ln\left(\rho \right)\;=\;ln\left(\sqrt{3}\;\cdot \;2\;\cdot \;u\right)$, deduced following the usual practice of invoking a uniform distribution in a type B evaluation of measurement standard uncertainty, *u*, for an instrument (test person) with finite resolution *ρ* [8]. This uncertainty determines how steeply the ogive Rasch curve (Figure 2) transitions from one score to another for the present case of dichotomous responses.

Further terms which could be added to the overall entropy of the measurement process (Figure 1) could describe, for example, the ability of the operator (*C*, clinician, in Figure 1) to make a diagnosis of a test person, as often analyzed in terms of area under the curve (AUC) of receiver operating curves (ROC), which can also be treated with the Rasch model [12,30]. In fact, entropy expressions of these various kinds can be added at will to the general Equation (2) describing the measurement process. Examples of such expressions are of the distortion, fuzziness, and lack of clarity of a wide range of attributes using basic measures of information content. These effects can be described, not only with probability theory, but even in terms of possibility, plausibility, belief and so on [6,12,35,36,37]. Information content can range from basic messages, such as the number of elementary symbols, to increasingly more sophisticated information, through syntax, semantic and pragmatic aspects of meaningful information in many contexts. Entropy can be a semantic measure based on meaning rather than just the syntax and as such supports not only descriptions of uncertainty, but also the identification of metrological references for comparability: the best units for traceability are those with the most order (i.e., least entropy, as an example of the principle of least action [12]—as discussed in the next section).

In summary, informational entropy (by analogy with thermodynamic entropy as ‘useful energy’ [13]) is a key concept when seeking metrological quality-assurance of measurement systems, both in terms of measurement uncertainty and of traceability.

### 2.4. Entropy and Construct Specification Equations

Richard Feynman [38] wrote on his last classroom blackboard: “What I cannot create, I do not understand”. By being able to create and understand, and in turn claim the highest level of validity, a CSE provides a more specific, causal, and rigorously mathematical conceptualization of item attributes (e.g., memory task difficulties) than any other kind of construct theory (Section 1). Indeed, as indicated in Figure 1, in principle, constructs can be associated with each element (i.e., object, instrument, operator, etc.) of a measurement system [20]. CSEs were first introduced in 1982 by Stenner and Smith [10] when explaining item (task) difficulty for the memory test Knox Cube Test (KCT) [39] focusing mainly syntax-based explanatory variables, such as the number of blocks tapped. Their work has contributed to the field of education science with the Lexile measure of reading ability as a prominent example [21,40]. However, a comparable effort in health care has been slow to develop.

The focus in this paper is on the formulation and verification the CSEs for memory task difficulties, that is, the construct attributed to the measurement object (entity *A* in Figure 1). The relationship between empirically measured memory task difficulties illustrated in this paper can be expressed in terms of the ‘something’ (i.e., the explanatory variables), which explains (or “causes”) why some memory tasks are easier than others. The formulation of a CSE (for the construct memory task difficulty) is often defined as a linear combination of a set, *k*, of explanatory (independent) variables, **X**:(4)$$\hat{Y}=\underset{k}{\sum }\beta_{k}\;\cdot \;x_{k}$$

State-of-the-art multivariate formulation of CSEs includes three steps of a principal component regression (PCR) [12]: (i) Principal component analysis (PCA) amongst the set of explanatory variables, **X*_k_***; (ii) linear regression of the measured memory task difficulty values *δ_j_* (or person memory ability values *θ_j_*) against **X’** in terms of the principal components, **P**; and (iii) conversion back from principal components to the explanatory variables, **X*_k_***, in order to express the CSE for the item (or person) attribute. The results of this PCR analysis reveal how much each explanatory variable contributes to explaining and predicting the observed variation in the attribute of interest.

Entropy, perhaps reflecting its semantic formulation, turns out to be a dominant factor in explaining task difficulty in these memory tests, as will be demonstrated experimentally in Section 3. It is expected that more ordered sequences will be easier to remember, i.e., the informational entropy is lower (i.e., more information) when the order of the test (e.g., sequence of blocks or digits) is greater, as previously considered for non-verbal tests of this kind [41,42]. It will be shown in Section 3 how much the explanatory variable *Entropy,* together with other terms—*Reversals* and *Average distance*—contribute to determining memory task difficulty in non-verbal memory tests such as CBT and DST. Furthermore, when regressing the empirical memory task difficulty values *δ_j_* against corresponding predicted values, *zR*, to yield the CSE, the goodness-of-fit *R*^2^ index indicates the amount of variance in task difficulty accounted for by the construct theory. Although Schnore and Partington [41] report appreciable entropy reductions from the symmetries of their stylized block tests, our current studies do not appear to offer any symmetry advantages when recalling for instance CBT sequences.

Continuing our presentation of entropy and memory task difficulty, the CBT and DST (and the Knox Cube Test (KCT), as shown in our earlier work [8,12,43], and other memory sequence tests) can be characterized in general in terms of a message in which a number, *N_j_*, (*j* = 1,…,*M*) of symbols of *M* different types (blocks or digits) can be distributed in a number, *G*, of categories (or cells) $G=\overset{M}{\underset{j=1}{\sum }}N_{j}$. The probability of encountering the *j*^th^ symbol is $p_{j}=\frac{N_{j}}{G}$, which can be summed to unity. The total number, *P*, of messages that can be obtained by distributing the symbols at random over the *G* cells (with never more than one symbol per cell) is $P=\frac{G!}{\prod_{j=1}^{M}N_{j}!}$. The information theoretical “Shannon” entropy (as invoked in Figure 1 and Section 2.3), which is a measure of the amount of information in these messages, is then given by the classic Brillouin expression [17]:(5)$$H\left(Z\right)\;=\;K\;\cdot \;lnP=K\;\cdot \;\left[ln\left(G!\right)-\overset{M}{\underset{j=1}{\sum }}ln\left(N_{j}!\right)\right]\;≅\;K\;\cdot \;\left[G\;\cdot \;ln\left(G\right)-\overset{M}{\underset{j=1}{\sum }}N_{j}\;\cdot \;ln\left(N_{j}\right)\right]$$
where *K* is an arbitrary constant. Stirling’s approximation of *ln*(*G*!) in the final terms of Equation (5) applies when *G* and *N* are large, but with modern computer power the approximation is no longer necessary when evaluating the factorial terms.

There remains today, however, some misconceptions about entropy, e.g., “As seen from (the Shannon) formula, the entropy of a string only depends on the relative proportions of symbols and is unaffected by the order in which they appear” [44]. This appears to ignore the work of Brillouin [19] whose expression (Equation (5)) for entropy can be evaluated for a number of sequences which not only have increasing numbers, G, of taps but also some repeats, N, of the same symbol (block or digit in the KCT, CBT or DST tests). Task difficulty, δ, would be expected, according to Equation (6) derived from Equation (5), to increase in proportion to $ln\left(G_{j}!\right)$, where G is the number of symbols encountered in a message. The second entropy term in this equation shows the expected decrease in entropy (and thus decreased task difficulty) from *N_j_* repeats:(6)$$\delta \propto Entropy=+\left[ln\left(G!\right)-\overset{M}{\underset{j=1}{\sum }}ln\left(N_{j}!\right)\right]$$

Recent examples of how this has been applied in memory tests is for the KCT are presented by Pendrill et al. [8], Pendrill [12] and Melin et al. [43]. These KCT results will be set in relation to the experimental demonstrations for the other memory tests CBT and DST in this paper in Section 3.3.

## 3. Results

This section presents: (i) how entropy can be used to explain memory task difficulty in CBT and DST; and (ii) how different items from different tests can be combined according to entropy-based criteria. Below, the CBT and DST cases are first handled separately and then related to each other and, finally, combined to create a new measure of memory with better reliability.

### 3.1. Case Study: Block Tapping Recall—Corsi Block Test

Table 1 shows the empirical memory task difficulty values, *δ,* and corresponding explanatory variables; *Entropy, Reversals* and *Average distance*, used for developing the CSE for CBT with PCR (Section 2.4). As there are no repetitions (*N* = 0) of blocks tapped in CBT, the term *Entropy* is evaluated using the Brillouin-expression (Equation (5)) as a function of solely the number, *G*, of blocks tapped. Applying the three steps in the PCR described in Section 2.4 yielded the following CSE: (7)$$zR_{j}=-7\left(3\right)\;+\;2.0\left(7\right)\;\times \;Entropy_{j}-\;0.8\left(1.2\right)\;\times \;Reversals_{j}-0.1\left(3\right)\;\times AveDistance_{j}$$
which is dominated by *Entropy* (Equation (6)) and where there are negligible contributions from the number of *Reversals* and from *Average distances* in each test sequence compared with the measurement uncertainties in the *β*-coefficients (Equation (4)) of the various explanatory terms in Equation (7) (given in parentheses with a coverage factor *k* = 2). The contributions from each of the explanatory variables are also shown graphically in Figure 3a. It is evident that *Entropy* increases as the empirical memory task difficulty, *δ,* increases*,* while *Reversals* and *Average distance* contribute just noise close to 0.

A regression of the empirical memory task difficulty values, *δ*, against corresponding estimated *zR* from the CSE, yielded a Pearson coefficient of *r*
$=\sqrt{R^{2}}$ = 0.98, indicating a high accuracy of the prediction. Furthermore, as shown in Figure 3b the empirical memory task difficulty values, *δ*, lie within the corridor of predicted uncertainties. i.e., *zR* ± *UzR* from the PCR (Section 2.4).

### 3.2. Case Study: Digit Recall—Digital Span Test

For the second memory test DST studied here, a summary of the empirical memory task difficulty values, *δ*, and corresponding explanatory variables used for developing a CSE is given in Table 2. The CSE for DST is found to be very similar to that demonstrated above for CBT (Equation (7)) with *Entropy* (from Equation (6)) as the dominate explanatory variable and with larger measurement uncertainties for the β-coefficients of the other two explanatory variables *Reversals* and *Average distance*:(8)$$zR_{j}=-10\left(3\right)\;+\;1.5\left(3\right)\;\times \;Entropy_{j}+\;0.2\left(4\right)\;\times \;Reversals_{j}-0.2\left(1.5\right)\;\times AveDistance_{j}$$

Further similarities between the CSE for DST and CBT are the noise around 0 from *Reversals* and *Average Distance* (Figure 4a); the high accuracy of the prediction (Pearson coefficient *r*
$=\sqrt{R^{2}}$ = 0.98); and the empirical memory task difficulty values, *δ*, found to be lying within the corridor of predicted uncertainties, i.e., *zR* ± *UzR* from the PCR (Figure 4b).

As for CBT, the DST sequences of the data did not include any repeated digits. However, another version of the DST (from the GBG MCI study [25]) does include repeats, for example a 9-digit sequence which has three digits repeated twice each. Such a sequence with duplicated digits (*N* = 2) should have higher order and convey more information (less entropy) compared with a digit sequence with the same number, *G*, of digits but with no duplicates, according to Equation (6). As an example, this 9-digit sequence is predicted to have an *Entropy* contribution and corresponding task difficulty comparable with those properties of an 8-digits sequence without repeats. This ability to indicate the equivalence of different test items—even those administered in different cohorts and cultural setting—adds to the theoretical justification for considering an information theoretical approach including *Entropy* as an explanatory variable when explaining memory task difficulty.

### 3.3. Contrasting CSEs from Knox Cube Test, Corsi Block Test, and Digit Span Test

As described above, the CBT is a block tapping test. The KCT is a similar block test but with only four blocks in a line instead of the nine spread in two dimensions in CBT. Task difficulty in KCT has been explained by Stenner and his colleagues as a function of *Number of taps*, *Reversals*, and *Sequence length*. Importantly, they did not adopt an entropy-based approach. Alternatively, in our recent work, we explored an entropy-based argumentation to investigate the advantages CSE for memory task difficulty in the KCT based on Equation (6). i.e., where repeated taps occur [8,12,43]. Pendrill [12] presented the following CSE:(9)$$zR_{j}=-9\left(5\right)\;+\;2\left(1\right)\;\times \;Entropy_{j}\;+\;0.8\left(1.4\right)\;\times \;Reversals_{j}+0.7\left(2.9\right)\;\times AveDistance_{j}$$

Despite the different layouts of these three non-verbal memory recall tests, one can note that the entropy terms in the CSEs from Equation (7) (CBT) and Equation (8) (DST) for memory task difficulty are essentially equivalent to Equation (9) (KCT), within the quoted measurement uncertainties. This provides further support for an information theoretical approach with *Entropy* as an explanatory variable in CSEs for predicting the memory task difficulty of non-verbal test sequences.

### 3.4. Combination of Memory Items from Corsi Block Test and Digit Span Test

The similarities in the administration of the two memory tests CBT and DST, as well as their similar CSEs, invite the novel combination of the separate items into one set of memory tasks. The main advantage of combining block and digit items in this way is that it provides more information about the test persons which in turn, leads to reduced measurement uncertainties (better reliability) in the assessed persons’ memory abilities. Combining tests in this way, with judicious choice of items according to an entropy-based criterion for equivalence, leading to more efficient testing. This improves not only the number of degrees of freedom but also the targeting of the memory test, where the span of person abilities should match—both in terms of centering, width, and coverage (without gaps)—the corresponding range of task difficulty (Section 2.2). This is shown in Figure 5 where uncertainties, U(θ), for each person’s memory ability are reduced in the new combined test by approximately 0.5 logits compared with U(θ) derived for only CBT. That reduction in measurement uncertainties corresponds to an improvement in reliability from 0.66 for only CBT to 0.70 for the combined memory test.

In addition, combining items from CBT and DST provides, firstly, a hierarchical ordering of the different memory tasks, and secondly, numerical values to compare the difficulty of different memory tasks. Figure 6 show a histogram of the memory items, CBT in blue and DST in grey, where the two easiest DST memory tasks, i.e., 3-digit sequences, are approximately 2 logits easier than corresponding 3-tapping sequences for CBT. In general, the CBT memory tasks are somewhat more challenging than DST, even though the two sets of sequence tests have roughly the same number of entities i.e., lengths of sequence. However, with the same length (and without duplicates), the predicted task difficulty *zR* for item *j* would be the same if *Entropy* had been the only explanatory variable, assuming those 3-taps/digits sequences have negligibly small contributions from the two other explanatory variables, *Reversals* and *Average distance* (Table 1 and Table 2), as demonstrated experimentally (Section 3.1 and Section 3.2).

## 4. Discussion

An information theory approach has been adopted when explaining the difficulty of the task of recalling non-verbal sequences (of blocks and digits) in memory tests. We hypothesize entropy is a measure of order; the higher the order, the lower the entropy, and a more ordered task will generally be easier to perform. We have demonstrated experimentally: (i) how entropy can be used to explain memory task difficulty; and (ii) how different items from different tests can be combined according to entropy-based criteria.

For this purpose, raw data from experiment had first to be transformed using the Rasch model to provide separate and quantitative estimates of task difficulty and person (instrument) ability on a conjoint interval scale. An entropy-based explanation of the Rasch model has been given in Section 2.3 which relates measurement system response to task difficulty, person (instrument) ability and discrimination. The Rasch model enables the establishment of metrological references for counted fraction data in terms of measurement system analysis (MSA) and the accuracy of classification of categorical data. Furthermore, from the metrological point of view, when establishing objective and scalable metrological units, CSEs come close to ‘recipes for certified reference materials’ for traceability, as in chemistry [5,43], emphasizing their essential role in both verification and validation, for instance in the present case when establishing an item bank of memory test item difficulty [12].

Our understanding means that memory task difficulty could be sufficiently explained and captured by mathematically formulating CSEs where entropy is shown to be the dominant explanatory variable. Strikingly, for the memory tests CBT and DST, as well as in our previous work on KCT [8,12,43], it is found that essentially the same CSE describes task difficulty within uncertainties for all these non-verbal sequence tests, with basically just one explanatory variable, namely *Entropy*. These results contrast with the conclusions of several previous studies, e.g., [10,41,42,45] where more than one explanatory variable was claimed to be significant, while at the same time not explicitly declaring measurement uncertainties nor using the Rasch model.

Additional novelties in the current work are associated with using the classic Brillouin [19] entropy expression to explain the difficulty of recalling non-verbal sequences in memory tests (to include both CBT and DST): (i) the entropy associated with the number of taps/digits (Equations (5) and (6)) is a semantic construct communicating meaning, which differs from the corresponding relation postulated by others in their early work on KCT [10,45]; (ii) in particular, Equation (6) models the reduction in entropy (and hence reduced memory task difficulty) associated with tasks involving duplicate symbols (such as repeated digits in a DST sequence); and (iii) this opens up opportunities to explore the significance of the degree of order in terms of entropy, which should reasonably be applicable to other neuropsychological tests of cognition.

As demonstrated in this paper, entropy-based CSEs can show the equivalence of tasks (with different sequences) in formerly separate versions of these non-verbal tests in a way not studied by others, to our knowledge. That in turn enables different tests to be merged, thereby increasing the number of items in a new composite test as well as filling in the “gaps” in the measurement scale by judicious choice of items, as illustrated in Figure 6. Both aspects—increased degrees of freedom and better coverage—belong to the advantages of formulating metrological reference materials semantically in what can be regarded as pioneering an extension of metrological principles to human-based measurements. These advances lead to improved reliability (reduced uncertainties) in person ability estimates (as shown in Figure 5). In addition to the focus on memory task difficulty in this study, CSEs can also be associated with each element (object, instrument, operator, etc.) of a measurement system [18].

On-going work on CSEs for person memory ability (as the ‘instrument’ of the measurement system), which are an additional output of analysis using the Rasch model of the memory test data, is under study in the NeuroMET research project [24]. So far, well-known dementia-related biomarkers, i.e., physical and chemical quantities, for each person in the cohort have been tested as explanatory variables [20]. As mentioned in the Introduction, not only task difficulty but also person ability can be explained semantically with the entropy concept, where inter-connectiveness in the brain is a ‘major biomarker’ for cognitive ability [46]. This suggests that future studies should focus on structural and functional brain networks, with continuing emphasis on the concept of entropy.

Moreover, the use of entropy as an explanatory variable in measurements goes beyond neuropsychological assessments of cognition. The concept of entropy can also explain efficiency of any organization in terms of entropy-based measures of synergy [47] or how well human muscles can deliver force ergonomically, for instance when climbing stairs or lifting heavy packages [48]. This brings us back and links our work to the basic underlying principle of entropy in thermodynamics. As coined by Carnot [13], who was concerned that “In any machine the accelerations and shocks of the moving parts represent losses of *moment of activity*… In other words, in any natural process there exists an inherent tendency towards the dissipation of useful energy.” Thus, apart from the rattling machines of the early Industrial Revolution, one can consider how well any task of a certain difficulty is performed by an agent of a certain ability in terms of entropy.

There are some limitations to bear in mind when interpreting the findings of the methodological development presented in this paper. Firstly, CBT, DST and KCT were purposively chosen as they are non-verbal memory tests and rather straightforward to administer. Thus, additional work is required for other memory tests, such as word recalling sequences, to further explore the significant role of entropy in explaining memory task difficulty. Secondly, as is evident from Figure 3b and Figure 4b, the “corridor” of PCR modeling uncertainties, UzR, is (much) wider than the observed memory task difficulty, δ, with its measurement uncertainties, respectively. The measurement uncertainties for each memory task’s difficulty, u(δ) propagate through the PCR, and, in turn the u(δ) have implications for U(β) and UzR together with uncertainties in the fit itself, which is an issue of sample size, collinearity and measurement disturbance. However, we mainly interpret this as indicating that there are sources of dispersion when making the multivariate regression which are not yet accounted for. Finally, we focused not on the mental processes in the brain of each test person (which are more related to explaining person ability, θ) considered by previous workers, but rather on the aspects of the perceived object, that is, factors which can explain task difficulty, δ. But as suggested above, forthcoming work are expected to address structural and functional brain networks, with an emphasis the concept of entropy also for mental processes.

## 5. Conclusions

As part of our research to provide descriptions of the intrinsic properties of a measurement system, where a test person acts as a measurement instrument, the use of entropy as a powerful explanatory variable has been found to explain the difficulty of recalling non-verbal sequences in memory tests.

Entropy-based CSEs can show the equivalence of tasks (with different sequences of blocks and numbers) in formerly separate versions of these three—CBT, DST and KCT—non-verbal memory tests. Despite the different layouts of these three tests, one can note that the entropy terms in the CSEs are essentially equivalent, within the quoted measurement uncertainties. Equal values of entropy indicate equivalence of different items, and this is a key—together with the conceptually understanding and design of those short-term memory tests—to combine items to form new (and more efficient) tests.

In turn, the resulting improved validity of memory tests accruing from entropy-based CSEs and the design of novel, combined tests will contribute to ensuring reliable diagnosis of disease such as dementia by insisting on proper scoring, exploiting the full potential of the concept of entropy in explaining semantically memory task difficulty and by estimating conjointly the memory ability of each cohort member. Moreover, this will hopefully reduce the risks of incorrect interpretation of the degree of correlation between cognition and various biomarker data, which otherwise would mean potential risks of incorrect monitoring and improper treatment of neurodegenerative diseases.

## Figures and Tables

**Figure 1 entropy-23-00212-f001:**
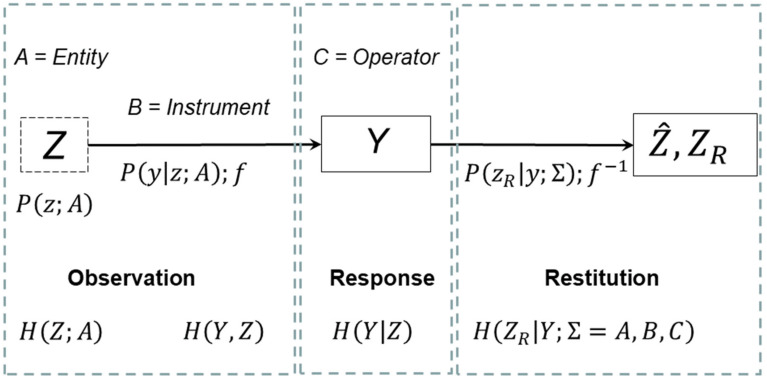
Probabilistic and entropy models of the measurement system and processes (inspired, in part, by the probabilistic model of Rossi [6] Figure 5.5). *Z*—measurand; *Y*—response; *R*—restitution; *P*—probability, and *H*—entropy. The neuropsychological assessment of cognition is made in terms of a metrological measurement system analysis (MSA), where the cognitive task is the measurement object (entity *A*), the human responder (patient whose memory is tested) is the measurement instrument (entity *B*) and the test is administered by an operator, usually a clinician (entity *C*) [7]. The concept of entropy (*H*) is invoked to aid explanation of how information is lost, distorted or gained throughout the measurement process from observation of the measurand (*Z*), via response (*Y*) to restitution (*Z_R_*) [8].

**Figure 2 entropy-23-00212-f002:**
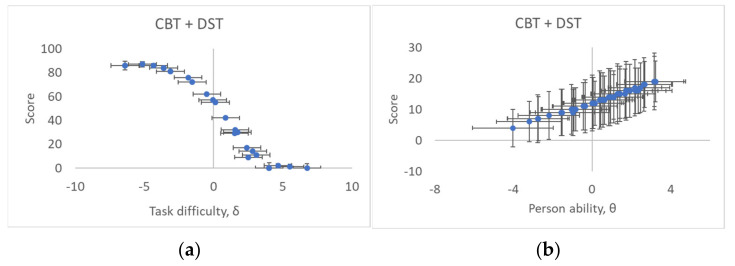
Observed S- or ogive-curve correlating observed responses with the Rasch-estimated (**a**) item difficulty and (**b**) person ability for the combined test items of the Corsi Block Test (CBT) + Digital Span Test (DST), described in Section 3.4. Measurement uncertainties quoted with coverage factor *k* = 2.

**Figure 3 entropy-23-00212-f003:**
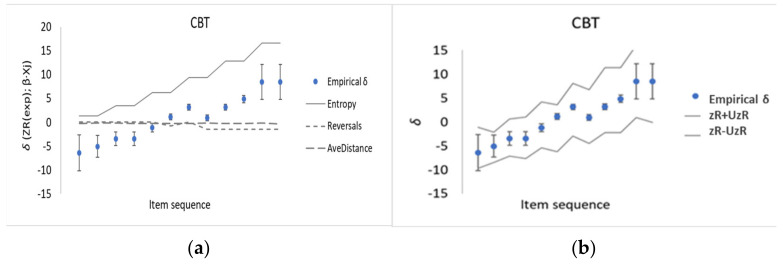
(**a**) Predicted contributions, Δ*δ*, to task difficulty from the three explanatory variables *Entropy* (Equation (6)), *Reversals* and *Average distance* for the CBT. (**b**) Dots with uncertainty intervals indicate the CBT empirical memory task difficulty, δ and corridors of modeled uncertainties show *zR* ± *UzR* (grey lines) for the predicted *zR* values from the PCR, coverage factor *k* = 2. For both (**a**) and (**b**) item sequences correspond to those given in Table 1.

**Figure 4 entropy-23-00212-f004:**
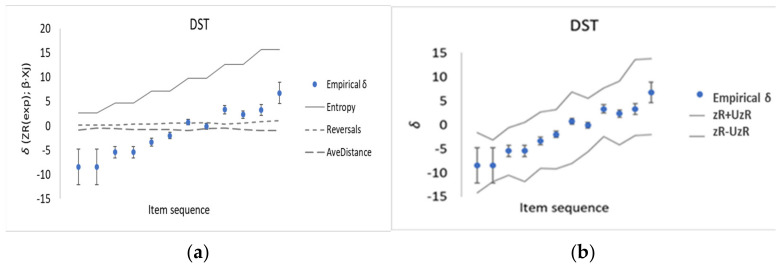
(**a**) Predicted contributions, Δ*δ*, to task difficulty from the three explanatory variables *Entropy* (Equation (6)), *Reversals* and *Average distance* for the DST. (**b**) Dots with uncertainty intervals shows the DST Empirical memory task difficulty, δ and corridors of modeled uncertainties shows *zR* + *UzR* (grey lines) for the predicted *zR* values from the PCR, coverage factor *k* = 2. For both (**a**) and (**b**), item sequences correspond to those given in Table 2.

**Figure 5 entropy-23-00212-f005:**
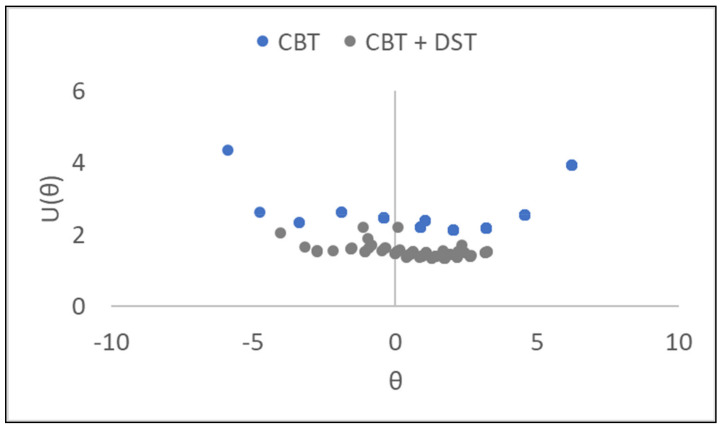
Measurement uncertainties (U(θ), *y*-axis) for each person’s memory ability from CBT in blue and from the combined CBT and DST in grey. On the *x*-axis, ability θ ranges, left to right, from least able persons to most able persons. Measurement uncertainties with coverage factor *k* = 2.

**Figure 6 entropy-23-00212-f006:**
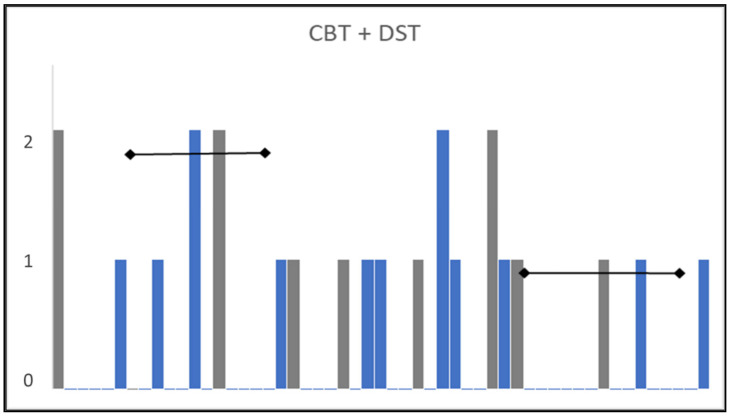
Histogram (occupancy on the *y*-axis) of empirical memory task difficulty, δ, (*x*-axis) for CBT in blue and DST in grey scaled together from left to right for the easiest tasks to most challenging tasks. Item order corresponds to Table 1 for CBT and Table 2 for DST. Horizontal error bars indicate measurement uncertainties with coverage factor *k* = 2.

**Table 1 entropy-23-00212-t001:** The empirical Rasch estimates, *δ_j_*, for each item, *j*, and its explanatory variables in CBT.

Tapping Sequence	Empirical Memory Task Difficulty, δ	U(δ) *k* = 2	Entropy (Equation (6))	Reversals	Average Distance
2 taps, 1st	−6.5	3.8	0.69	0	10.5
2 taps, 2nd	−5.1	2.3	0.69	0	4.7
3 taps, 1st	−3.5	1.5	1.79	0	7.7
3 taps, 2nd	−3.5	1.5	1.79	0	9.9
4 taps, 1st	−1.2	0.8	3.18	0	10.3
4 taps, 2nd	1.1	0.6	3.18	1	9.9
5 taps, 1st	3.1	0.6	4.79	0	10.4
5 taps, 2nd	0.9	0.6	4.79	2	7.5
6 taps, 1st	3.1	0.6	6.58	2	9.8
6 taps, 2nd	4.8	0.8	6.58	2	9.7
7 taps, 1st	8.5	3.7	8.53	2	6.1
7 taps, 2nd	8.5	3.7	8.53	2	13.3

**Table 2 entropy-23-00212-t002:** The empirical Rasch estimates, *δ_j_*, for each item, *j*, and its explanatory variables in DST.

Number Sequence	Empirical Memory Task Difficulty, δ	U(δ) *k* = 2	Entropy Equation (6)	Reversals	Average Distance
3 digits, 1st	−8.5	3.7	1.79	1	3.7
3 digits, 2nd	−8.5	3.7	1.79	1	2.0
4 digits, 1st	−5.4	1.2	3.18	1	2.5
4 digits, 2nd	−5.4	1.2	3.18	2	3.5
5 digits, 1st	−3.4	0.8	4.79	2	3.2
5 digits, 2nd	−2.0	0.7	4.79	3	3.4
6 digits, 1st	0.8	0.6	6.58	3	4.3
6 digits, 2nd	−0.1	0.6	6.58	4	2.7
7 digits, 1st	3.3	0.9	8.53	2	2.1
7 digits, 2nd	2.3	0.8	8.53	3	3.6
8 digits, 1st	3.3	1.1	10.60	5	4.4
8 digits, 2nd	6.8	2.2	10.60	6	4.3

## Data Availability

Part of the data presented in this study are available on request from the corresponding author. The data are not publicly available due to ongoing longitudinal data collection.

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
