# Peer review of "The Role of Entropy in Construct Specification Equations (CSE) to Improve the Validity of Memory Tests"

_entropy, 2021, doi:10.3390/e23020212_

Round 1

Reviewer 1 Report

The paper is very well written. Since I am only an expert in
information theory and not in Construct Specification Equations or
Memory Tests I will restrict my comments to
those parts of the paper.

The parallel discussion of thermodynamic entropy and information
theory is fine, in principle. But I want to caution a superficial  that
there is a whole literature that debates the thermodynamic meaning
of information entropy. O. Maroney's work might be of interest to
the authors. A starting point could be Maroney's Stanford
encyclopedia entry 'Information Processing and Thermodynamic
Entropy'. I don't suggest the authors should add much here, but

The second sentence in the last paragraph on page three ("The
amount of ...") is wrong
the way it is written. The explanation afterwards is correct; I
recommend reformulating that sentence.

The entropy treatment is based on Figure 1. I find that Figure
rather confusing. The same is true for Equation 2 and the
surrounding paragraph. The two are connected. The notation mixes probability
distributions and random variables. It is better to write Shannon
entropy in terms of random variables when conditional entropies
are involved. Rewriting Eq (2) using the information on P and Q
gives H(Q | P) = H(Y | Z | Z) which is nonsensical. If, instead,
only Y and Z are used, then H(Q | P) = H(Y | Z) = H(Q). Thus, the
notation needs some cleaning up, I believe.

Furthermore, the relation between Q and P and A and B etc is
unclear. Why is H(P) = H(Z | A) when P = P(z)? Figure 1 needs more explanation of the various variables to help the reader.

Since I was stuck in the initial parts pf the
methodology I could not follow all of Section 2.4. But I wonder
what the purpose of Section 2.3 is when in Section 2.4 the Hartley
entropy is used (which assigns equal probability to all possible
events), Eq (5).

I am afraid I cannot give a holistic view of the paper or its
potential impact. The authors have the difficult task to bridge
the gap between two disconnected communities. The paper is very
well written, but I recommend to make sure both communities can
understand enough to be able to follow the arguments.

Author Response

Referee 1: The parallel discussion of thermodynamic entropy and information theory is fine, in principle. But I want to caution a superficial that there is a whole literature that debates the thermodynamic meaning of information entropy. O. Maroney's work might be of interest to the authors. A starting point could be Maroney's Stanford encyclopedia entry 'Information Processing and Thermodynamic Entropy'. I don't suggest the authors should add much here, but…

The second sentence in the last paragraph on page three ("The amount of ...") is wrong the way it is written. The explanation afterwards is correct; I recommend reformulating that sentence.

Authors’ response: Text in the last paragraph on page two and first at page three has been modified and some words and references have been added, as requested (yellow highlighted).

Referee 1: The entropy treatment is based on Figure 1. I find that Figure rather confusing. The same is true for Equation 2 and the surrounding paragraph. The two are connected. The notation mixes probability distributions and random variables. It is better to write Shannon entropy in terms of random variables when conditional entropies are involved. Rewriting Eq (2) using the information on P and Q gives H(Q | P) = H(Y | Z | Z) which is nonsensical. If, instead, only Y and Z are used, then H(Q | P) = H(Y | Z) = H(Q). Thus, the notation needs some cleaning up, I believe.

Furthermore, the relation between Q and P and A and B etc is unclear. Why is H(P) = H(Z | A) when P = P(z)? Figure 1 needs more explanation of the various variables to help the reader.

Authors’ response: Figure 1 and the mathematical notation have been corrected, as suggested. Instead of using the conditional notation when specifying the entity, we now use “;” viz.  and , when referring to a variable (Z) associated with an entity (A). See yellow highlights in section 2.3. at page five.

Referee 1: Since I was stuck in the initial parts pf the methodology I could not follow all of Section 2.4. But I wonder what the purpose of Section 2.3 is when in Section 2.4 the Hartley entropy is used (which assigns equal probability to all possible events), Eq (5).

Authors’ reply: The text in Section 2.4 has been modified to clarify (see yellow highlights): Brillouin’s expression (eq. 5) and our current applications are based on the same probabilistic entropy (à la Shannon) as referred to in Section 2.3. (Hartley entropy – not invoked here - is a more primitive form of entropy which does not need mention of probability theory – see reference [35].)

I am afraid I cannot give a holistic view of the paper or its potential impact. The authors have the difficult task to bridge the gap between two disconnected communities. The paper is very well written, but I recommend to make sure both communities can understand enough to be able to follow the arguments.

Authors’ response: We have rewritten the texts and provided further explanation of notation to help reader understanding, in particular see yellow highlights in section 2.3 and section 4.

Reviewer 2 Report

Authors should formulate the conclusions presented in more detail the contribution of the paper in relation to the works cited in the references.

In the conclusion section, the authors are lacking to justify in more detail the contribution of the research in relation to the similar works cited in the paper, citing in relation to which paper they are referring to. In the current form, the conclusion section does not present the contribution of this research in relation to results already published in other works.

Author Response

Referee 2:

Authors should formulate the conclusions presented in more detail the contribution of the paper in relation to the works cited in the References.

In the conclusion section, the authors are lacking to justify in more detail the contribution of the research in relation to the similar works cited in the paper, citing in relation to which paper they are referring to. In the current form, the conclusion section does not present the contribution of this research in relation to results already published in other works.

Authors’ reply:

Most claims to originality are made at present in the section 4. Discussion instead of in the Conclusions. Although some modifications are made in the Conclusions to emphasize key contribution from this work, see yellow highlights.